# Detraque: Dynamic execution tracing techniques for automatic fault localization of hardware design code

**Jiang Wu** [1], **Zhuo Zhang** [2]*, **Jianjun Xu** [1], **Jiayu He** [1], **Xiaoguang Mao** [1], **Xiankai Meng** [3], **Panpan Li** [1]

**1** College of Computer, National University of Defense Technology, Changsha, Hunan, China, **2** School of Information Technology and Engineering, Guangzhou College of Commerce, Guangzhou, Guangdong, China, **3** College of Computer and Information Engineering, Shanghai Polytechnic University, Shanghai, China

* zz8477@126.com

**Data Availability Statement:** https://github.com/wndif/Benchmark_Detraque.git.

**Funding:** The author(s) received no specific funding for this work.

## Abstract

In an error-prone development process, the ability to localize faults is a crucial one. Generally speaking, detecting and repairing errant behavior at an early stage of the development cycle considerably reduces costs and development time. The debugging of the Verilog program takes much time to read the waveform and capture the signal, and in many cases, problem-solving relies heavily on experienced developers. Most existing Verilog fault localization methods utilize the static analysis method to find faults. However, using static analysis methods exclusively may result in some types of faults being inevitably ignored. The use of dynamic analysis could help resolve this issue. Accordingly, in this work, we propose a new fault localization approach for Verilog, named Detraque. After obtaining dynamic execution through test cases, Detraque traces these executions to localize faults; subsequently, it can determine the likelihood of any Verilog statement being faulty and sort the statements in descending order by suspicion score. Through conducting empirical research on real Verilog programs with 61 faulty versions, Detraque can achieve an *EXAM* score of 18.3%. Thus, Detraque is verified as able to improve Verilog fault localization effectiveness when used as a supplement to static analysis methods.

## Introduction

HDL (Hardware Description Language) uses formal methods to describe digital circuits and systems. According to research, at present, more than 90% of ASICs and FPGAs in the United States' Silicon Valley is designed using HDL [1]. Recent increases in the complexity of hardware designs has challenged the ability of developers to identify defects in circuit descriptions. The need to ensure the functional correctness of modern electronic designs adds a layer of significant complexity to the development process, with verification and debugging often accounting for up to 70% of the design cycle [2]. The verification stage is intended to ensure that a design behaves as intended, while debugging focuses on localizing and correcting the

**Competing interests:** The authors have declared that no competing interests exist.

root causes of verification failures. Usually, developers detect faults in hardware designs by employing verification tools, such as model checkers [3] or by using test cases together with simulators and waveform trace tools. In both cases, the hardware designer identifies test vectors that lead to unexpected results. It is critical to utilize these tools when locating the faulty statements, as they reduce development time and overall project costs.

While previous works have attempted to address this problem, these works also exhibit certain shortcomings. For instance, some extant techniques automatically localize defects in the design source code, but suffer from high false positive rates [4]. In order to conduct design verification, some automatic error diagnosis and correction approaches require formal specifications [5], which typically do not scale to large designs. This paper aims to address a key related concern: as the current research on HDL fault localization shows, the most HDLs fault localization methods utilized today are based on static analysis (e.g., model-based methods). When using traditional static analysis diagnosis, it is challenging to locate arithmetic symbol faults or conditional statement faults [6–8]. By contrast, dynamic analysis mainly utilizes coverage information of test cases, and can therefore deal with various types of faults [9]. Compared with other model-based fault localization methods, value-based models require more calculations and are suitable only for small programs [10]; the average number of statements currently used by value-based models (ISCAS'85 and ISCAS'89) does not exceed 100 lines [11]. The dynamic analysis method has relatively loose requirements regarding the number of statements. Analysis of the existing research suggests that a more effective dynamic HDL fault localization method is required to supplement the previous static analysis methods.

While there are many similarities between HDLs and software languages, there are also many differences. Specifically, two key differences are highlighted. (1) Software programs are typically based around a serial execution model, where one line of code executes before the next. By contrast, HDL designs are inherently parallel and often include non-sequential statements, since separate portions of hardware can operate simultaneously. Whether the dynamic analysis method applicable in software languages can be applied to HDLs is one of the focuses of this paper. (2) Software programs often use test cases to evaluate functional correctness, such that individual test cases may pass or fail depending on the quality of the software. HDL designs, on the other hand, use the testbenches-programs with documented and repeatable sets of stimuli, to simulate the behaviors of a device under test (DUT). In this article, our approach attempt to combine a wealth of test cases with testbenches to create a fault localization method based on dynamic information that is suitable for HDL.

Thus, this paper proposes a new fault localization approach, Detraque, for use with Verilog. Currently, Verilog has a wide range of applications [12], a concise and efficient code style, and stronger logic gate-level and RTL (register transfer level) circuit description capabilities. This paper focuses on RTL Verilog to carry out research on HDL fault localization. In more detail, Detraque performs fault localization by tracing the dynamic execution information of programs. It first obtains the coverage information of the test cases from the program's dynamic execution information, then calculates the so-called suspicion value of different statements: the higher the suspicion value, the greater the likelihood that the statement is buggy. To evaluate our approach, Detraque is applied to seven programs obtained from real applications running on FPGA from OpenCores. The experimental results show that Detraque improves fault localization effectiveness on Verilog; furthermore, the findings reflect the feasibility and advantages of dynamic analysis technology in the context of Verilog fault localization.

The main contributions of this paper can be summarized as follows:

- We propose a new fault localization approach, Detraque, for Verilog. By obtaining dynamic execution through test cases, Detraque traces these executions to localize faults, this new

method can present the suspicion values of Verilog statements (i.e., their likelihood of being faulty) in descending order.

- We construct a test suite based on RTL Verilog programs obtained from real applications, and use fault injection technology to generate different single-fault versions. Compared with the traditional FPGA test suite, our proposed test suite can more effectively evaluate the fault localization ability. We shared the test suite on Github (https://github.com/wndif/Benchmark_Detraque.git).

- We conduct an experimental study on seven real-life programs with seven state-of-the-art evaluation metrics. Qualitative analysis of the experimental results shows that Detraque is effective at improving fault localization on Verilog.

## Background

As hardware systems become more complex, versatile, and ubiquitous, verification has emerged as one of the biggest bottlenecks in their design cycle [13]. While checking for the desired behavior is one side of the verification coin, debugging to find the root causes of faults is the other. First and foremost, the debugging functional faults in RTL design is widely accepted as one of the "pain points" of verification. Debugging of a single fault can take several weeks. Once a fault is detected, moreover, it needs to be classified as caused by either the design or the verification environment, a process that involves various assertions, monitors, checkers, etc. Isolating the cause is frequently difficult even at this coarse-grained level. Massive industrial-scale designs comprise several GB of simulation trace data and hundreds of thousands of lines of RTL source code; identifying the root cause of a fault within this vast corpus is comparable to finding a needle in a haystack. Notably, localization of the fault to any extent is highly valuable and could result in significant cost reductions. State-of-the-art debugging tools aid with visualization and "what-if" scenarios, but do not provide any localization of the root cause.

Automated fault localization for HDLs has been researched for decades. Researchers have earlier described a diagnosis tool for VHDL that employs the so-called "functional fault" models and reasons from first principles by means of constraint suspension [14–16]. Huang et al. [17] were the first to propose symbolic and simulation-based fault localization approaches for HDLs, which have been adopted by most of the subsequent research. Model-based fault diagnosis technology which has been used in hardware for a long time [18], aims to infer the possible location of the fault through the diagnostic model and program behavior observation results. Subsequently, Peischl et al. proposed a series of model-based fault diagnosis methods and tested them on both VHDL and Verilog [19, 20], optimizing the test suite and the model generation algorithm. In current research on HDL fault localization technology, model-based fault localization is the mainstream [13]. When using model-based diagnosis, it is assumed that a correct model is available for each program being diagnosed; that is, these models are expected to serve as the oracles of the corresponding programs. Differences between the behaviors of a model and the actual observed behaviors of the program are used to help find bugs in the program [10]. This approach does not require a large number of test cases and statistical information, or any restrictions on code size.

In recent years, several new fault localization methods for HDL have been developed. Bhattacharjee et al. [21] proposed an efficient methodology for automatically localizing design errors across design versions. The proposed technique, EvoDeb, can be easily integrated into a hardware configuration management framework and is scalable for large designs. However, it is targeted at specific version change bugs in the context of these designs. Rajashekar et al. [22]

use machine learning methods to aid fault localization in HDL; due to the small dataset size, however, the desired experimental results were not achieved. Mahzoon et al. [23] proposed a method based on symbolic computer algebra that is effective for large-scale and less complex programs; notably, while symbolic approaches are accurate, they suffer from combinatorial explosion.

In fact, the above HDL fault localization methods rarely make use of a program's dynamic execution information. It is necessary to detail the execution information of a program from certain perspectives, such as execution information for conditional branches or loop-free intraprocedural paths. This approach can be used to track program behavior; when the execution fails, such information can be used to identify suspicious code that is responsible for the failure. Code coverage indicates which parts of the program under testing have been covered during an execution. Using this information, it is possible to identify which components were involved in a failure, narrowing the search for the faulty component that caused the execution to fail. While technical methods based on program dynamic execution information are very commonly used in the field of software fault localization [10], they are rarely used in the field of HDL fault localization. Pal et al. [24] attempted to trace the source of failed test cases. Their approach employs a combination of dynamic execution and the symbolic approach, although the risk of combinatorial explosion from the symbolic approach still exists.

## Detraque approach

The workflow of Detraque method is shown in Fig 1. Detraque runs a large number of test cases on the test suite. Due to the unique characteristics of HDLs, different testbenches for different programs are needed when running test cases. We further need to control the input parameters of test cases in the testbench based on circuit design. Detraque then obtains program coverage information based on dynamic execution and calculates the suspicion value of the statements with evaluation metrics. Finally, it outputs a ranking list of statements in descending order of suspiciousness. The workflow of Detraque is made up of two main parts, *Dynamic Execution Cycle* and *SVE Algorithm to Detraque*. The following content of this section will describe the Detraque workflow in detail.

### Dynamic execution cycle

This step of Detraque is a collection of data that provides a specific view of the program's dynamic behavior. Specifically, it contains coverage information pertaining to the entities of a program that were or were not executed during the execution of several test cases. These entities could be individual statements, basic blocks, branches, or larger regions such as functions; essentially, they can be considered equivalent to basic blocks, assuming normal termination. During the execution of each test case, coverage information is collected that indicates whether or not the statements have been executed. Additionally, each test case is classified as either a passed test case or a failed test case.

The entire dynamic execution cycle in Detraque consists of three modules: *Test Bench*, *Test Suite*, and *Dynamic Execution*. In Verilog, the testbench is used to control the execution of the program. For each program to be tested, Detraque puts all test cases in a testbench and execute the program in a loop until each test case is executed (the test case here is the stimulus). After executing a test case, Detraque needs to clear the execution record stored in the testbench to avoid generating inaccurate coverage information. We named the collection of all test cases the *Test Pool*; the construction of the test pool will be explained in detail in Section.

The *Workspace* option of Modelsim provides the ability to obtain code coverage information. It can report multiple coverage situations such as the statement, branch, condition,

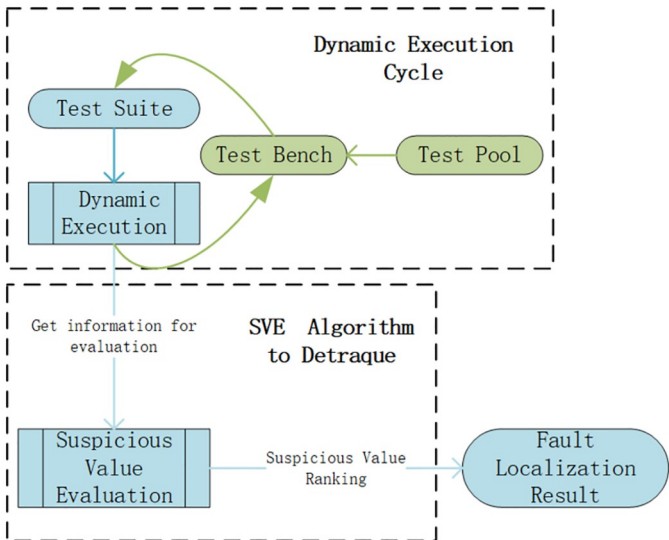

**Fig 1. Workflow of Detraque.** The workflow of Detraque is depicted in the figure. Divided into two main modules, one is the *Dynamic Execution Cycle*, the function is to execute the program through the *Test Bench* and collect the dynamic execution information of the program. The other part is the *SVE Algorithm to Detraque*, whose function is to calculate the suspicious value of the statement based on the information output in the *Dynamic Execution Cycle*. Finally, obtain the final fault localization result based on the suspicious value.

expression, and signal reversal. Detraque focuses primarily on the coverage of each statement. Detraque first writes all test cases into the testbench, then execute all test cases for each faulty version of the program using the command line, and subsequently collect the coverage information of the dynamic execution through the command line. Due to a large number of test cases, Detraque turns on the incremental compilation switch *incr* during the cycle to speed up the simulation process. Moreover, because Modelsim automatically saves the collected coverage information, it's important to clear this information after each simulation before beginning the next round of collection.

Detraque next needs to further organize the collected program coverage information and execution results. Detraque represents the collected information using the mathematical approach outlined below. Given a Verilog program $V$ with $Q$ statements, it is executed by a test suite $T$ with $P$ test cases, which contain at least one failed test case. As Fig 2 shows, the

$$
P \ testcases \begin{bmatrix} x_{11} & x_{12} & \cdots & x_{1Q} \\ x_{21} & x_{22} & \cdots & x_{2Q} \\ \vdots & \vdots & & \vdots \\ x_{P1} & x_{P2} & \cdots & x_{PQ} \end{bmatrix} \overset{errors}{\begin{bmatrix} e_1 \\ e_2 \\ \vdots \\ e_P \end{bmatrix}}
$$

$Q \ statements$

**Fig 2. The coverage and results of *P* executions.** Shown in the figure is a matrix based on program dynamic execution information. One dimension of the matrix is the executable statement $Q$, and the other dimension is the test case $T$. Each test case corresponding to an execution result is represented by an element $e_i$ in vector *error*. Each element in the matrix represents whether a certain test case executes a certain statement, which is distinguished by 1 and 0.

element $x_{ij} = 1$ means that the statement $j$ is executed by the test case $i$, while $x_{ij} = 0$ indicates otherwise. The $P \times Q$ matrix records the execution information of each statement in the test suite $T$. The error vector $e$ represents the test results. The element $e_i$ is equal to 1 if the test case $i$ failed, and 0 otherwise. The error vector shows the result of each test case (i.e., 1 for failure or 0 for non-failure).

## SVE algorithm to Detraque

The content of this section is divided into two parts. The first outlines the design of the SVE (suspicious value evaluation) algorithm to Detraque. The second contains a description of the parameters of the SVE algorithm, presented in the form of a case study.

**SVE algorithm design.** For each statement, four numbers are ultimately produced: specifically, the number of passed/failed test cases in which the statement was/was not executed. Adapting Abreuet al. [25], Detraque uses the notation $\langle \alpha_{np}, \alpha_{nf}, \alpha_{ep}, \alpha_{ef} \rangle$: here, the first part of the subscript indicates whether the statement was executed ($e$) or not ($n$), while the second indicates whether the test passed ($p$) or failed ($f$). For example, the $\alpha_{ep}$ of a statement denotes the number of tests passed and the fact that the statement was executed. The raw data is often presented as a matrix of numbers (binary in this case), with one row for each program statement and one column for each test case; here, each cell indicates whether a particular statement is executed (the value is 1) or not (the value is 0) for a particular test case. Additionally, there is a (binary) vector indicating the result (0 for pass and 1 for fail) of each test case. This data allows us to compute the $\alpha_{ij}$ values, where $i \in \{n, e\}$ and $j \in \{p, f\}$.

Applying a function that maps the four $\alpha_{ij}$ values to a single number (Detraque refers to these functions as evaluation metrics) for each statement allows us to rank the statements. Those with the highest metric values are considered the most likely to be buggy. Generally speaking, Detraque would expect buggy statements to have relatively high $\alpha_{ef}$ and relatively low $\alpha_{ep}$. When $\alpha_{ef}$ is maximal and $\alpha_{ep}$ is minimal (the statement is executed in all failed tests but no passed tests), all the metrics Detraque considers return maximal values. Notably, different metrics give different weights to passed and failed test cases, meaning that they generally result in different rankings.

**Algorithm 1** SVE Algorithm to Detraque.

```
1:
2: Input: Original program P = {s₁, s₂, s₃, ..., sₙ}. Test cases collec-
tion T = {t₁, t₂, t₃, ..., tₘ}.
3: Output: Descending suspicious statements list with suspicion values
RS.
4: Begin
5:   RS = {}; CoverageMetrix = 0; ErrorVector = 0
6:   Execution Trace = Run(P, T)
7:   for i = 1; i <= m; i++ do
8:     if (CorrectResult) do
9:       ErrorVector[i] = 0
10:     else do
11:       ErrorVector[i] = 1
12:     end if
13:     for j = 1; j <= n; j++ do
14:       if (sⱼ is executed by tᵢ) do
15:         CoverageMetrix[i][j] = 1
16:       else do
17:         CoverageMetrix[i][j] = 0
18:       endif
19:   for sⱼ in P do
20:       αₙₚ(sⱼ) = Σᵢ∈ₙₚ₍ₛⱼ₎ (1 - CoverageMetrix[i][j])
```

```
21:      α_ep(s_j) = Σ_{i∈ep(s_j)} CoverageMetrix[i][j]
22:      α_nf(s_j) = Σ_{i∈nf(s_j)} (1 - CoverageMetrix[i][j])
23:      α_ef(s_j) = Σ_{i∈ef(s_j)} CoverageMetrix[i][j]
24   switch(evaluation metrics type):
25:     case(type):
26:       Sus(s_j) = getSusvalue⟨α_np, α_nf, α_ep, α_ef⟩
27:     RS = RS ∪ {s_j, Sus(s_j)}
28:     Order (RS)
29   end for
30:   return RS
31: End
```

The detailed SVE algorithm is presented in 1. There are two inputs to the algorithm: the original program and the collection of test cases. The test cases here are sets of stimuli for the Verilog testbenches. Detraque saves a large number of test cases in the form of testbench stimuli. Statements such as *always*, *input*, and *output* are not considered in this paper when obtaining coverage information; these types of declaration statements do not affect the correct execution of the program and are considered correct by default. The output of the algorithm is a list of suspicious statements sorted by suspicion values *RS*. Detraque determines the execution result *ErrorVector* of the test cases and the *CoverageMatrix* of the statements according to the information of the dynamic execution.

Suppose that there is at least one failing test case in the original test suite. Line 5 initializes *RS*, *CoverageMatrix*, and *ErrorVector*. The structure of *CoverageMatrix* and *ErrorVector* is defined in Fig 2. Line 6 runs the program based on the designed test cases. Lines 7-18 assign values to *CoverageMatrix* and *ErrorVector* according to the dynamic execution information. In line 19-23, for each statement in the program, obtaining four parameters $\langle \alpha_{np}, \alpha_{nf}, \alpha_{ep}, \alpha_{ef} \rangle$. In lines 20-23, the summation subscript $i \in np(s_j)$ indicates the collection of test cases that have not executed statement $s_j$ and test passed; the summation subscript $i \in ep(s_j)$ indicates the collection of test cases that have executed statement $s_j$ and test passed; the summation subscript $i \in nf(s_j)$ indicates the collection of test cases that have not executed statement $s_j$ and test failed; the summation subscript $i \in ef(s_j)$ indicates the collection of test cases that have executed statement $s_j$ and test failed. Taking line 21 as an example, The value of $\alpha_{ep}(s_j)$ represents the sum of the number of test cases that have executed statement $s_j$ and test passed. For $\alpha_{np}(s_j)$ and $\alpha_{nf}(s_j)$ in lines 20 and 22, using $(1 - CoverageMatrix[i][j])$ to indicate that the statement is not executed. Lines 24-25 select different suspicion value evaluation metrics. Line 26 calculates the suspicion value for each statement. Line 27 joins the statement with a corresponding suspicion value to the list *RS*. Line 28 sorts the *RS*. Finally, Detraque obtains a sorted list of suspicion statements with suspicion values.

**Case study.** We present an example in Fig 3. The sample code snippet in the figure has a total of 15 lines, denoted by $S_1$ to $S_{15}$. The faulty statement is $S_7$; it is extracted from a real-world program. There are five test cases, which are represented by $t_1$ to $t_5$. The number below each statement in the figure corresponds to the execution of the statement. The value in the result column indicates that the program produces a correct result when test case $t_1$ to $t_4$ is executed, while the result is wrong when test case $t_5$ is executed.

Taking $S_2$ as an example, the notation $\langle \alpha_{np}, \alpha_{nf}, \alpha_{ep}, \alpha_{ef} \rangle$ is $\langle 1, 0, 3, 1 \rangle$. This means that there is 1 passed test case that does not execute $S_2$, 0 failed test cases that do not execute $S_2$, 3 passed test cases that execute $S_2$, and 1 failed test case that executes $S_2$. The last row, *rank*, shows the ranking of the suspicion value of statements after being calculated by evaluation metric GP03. The faulty statement $S_7$ is ranked first by GP03.

We have selected seven evaluation metrics [26] for calculating the suspicion value of statements. These methods are very commonly used in the field of suspicion value calculation and

| | Program V | | | | | | | | Bug information | | | | | Annotation | | |
|---|---|---|---|---|---|---|---|---|---|---|---|---|---|---|---|---|
| S1:align:<br>S2:begin<br>S3:if ($signed(a_e) > $signed(b_e)) begin<br>S4:b_e <= b_e + 1;<br>S5:b_m <= b_m >> 1;<br>S6:b_m[0] <= b_m[0] \| b_m[1];<br>S7:end else if ($signed(a_e) > $signed(b_e)) begin<br>S8:a_e <= a_e + 1; | | | | S9:a_e <= a_e + 1;<br>S10:a_m <= a_m >> 1;<br>S11:a_m[0] <= a_m[0] \| a_m[1];<br>S12:end else begin<br>S13:state <= add_0;<br>S14:end<br>S15:end | | | | S7 is faulty. Correct form:<br>S7:end else if ($signed(a_e) < $signed(b_e)) begin | | | | | (1)The evaluation metric we used is GP03;<br>(2)Take S2 as an example, $a_{np}$=1, $a_{nf}$=0, $a_{ep}$=3 and $a_{ef}$=1;<br>(3)Faulty statements S7 is ranked 1st. | | |

| test | S1 | S2 | S3 | S4 | S5 | S6 | S7 | S8 | S9 | S10 | S11 | S12 | S13 | S14 | S15 | RESULT |
|---|---|---|---|---|---|---|---|---|---|---|---|---|---|---|---|---|
| t1 | 0 | 1 | 1 | 0 | 1 | 1 | 0 | 0 | 0 | 0 | 1 | 0 | 1 | 0 | 1 | 0 |
| t2 | 0 | 0 | 0 | 0 | 0 | 0 | 0 | 0 | 0 | 0 | 0 | 0 | 0 | 0 | 0 | 0 |
| t3 | 0 | 1 | 1 | 0 | 1 | 0 | 1 | 1 | 0 | 0 | 1 | 0 | 1 | 0 | 1 | 0 |
| t4 | 0 | 1 | 0 | 0 | 1 | 0 | 1 | 1 | 1 | 0 | 1 | 1 | 1 | 0 | 1 | 0 |
| t5 | 0 | 1 | 1 | 1 | 1 | 1 | 1 | 0 | 0 | 0 | 1 | 0 | 1 | 0 | 1 | 1 |
| suspiciousness | 0.00 | 0.86 | 0.64 | 1.00 | 0.86 | 0.00 | 1.19 | 1.00 | 1.00 | 0.00 | 0.86 | 1.00 | 0.86 | 0.00 | 0.86 | GP03 |
| rank | 12 | 6 | 11 | 2 | 8 | 13 | 1 | 3 | 4 | 15 | 7 | 5 | 9 | 14 | 10 | |

**Fig 3. The lower case illustrating seven evaluation metrics of fault localization approaches.** The figure content is divided into upper and lower parts. The upper part includes the program, injected fault information, and annotation. The lower part is the matrix information based on Fig 2 composed of statements and test cases, S1 to S15 are executable statements of the program, and t1 to t5 are test cases.

achieve relatively good results. The purpose of the experimental setup is to verify whether these methods can have the same effect for Verilog fault localization(see Fig 3). All formulas are calculated using the above four parameters. Finally, a suspicion value is obtained to help debuggers distinguish between possible correct statements (lower suspicion value) and possible incorrect statements (higher suspicion value). During the suspicion value calculation process "noise" (the existence of which is strongly related to the wrong behavior but the correct code) and "weak signal" (that is, the wrong behavior is difficult to detect) are frequently observed; thus, Detraque chooses a variety of calculation methods, as shown in Table 1.

## Experimental study

### Experimental setup

In order to verify the effectiveness of Detraque on Verilog, we use seven evaluation metrics (see Table 1) to conduct our experiments. Detraque tracks the dynamic execution information of the subject program based on a large number of test cases. Detraque then utilizes the evaluation metrics listed in Table 1 to calculate the suspicion of the statements. The physical

**Table 1. Definitions of evaluation metrics used.**

| Name | Formula |
|---|---|
| Russel_Rao | $\dfrac{\alpha_{ef}}{\alpha_{ef}+\alpha_{nf}+\alpha_{ep}+\alpha_{np}}$ |
| Dstar | $\dfrac{\alpha_{ef}{}^{*}}{\alpha_{nf}+\alpha_{ep}}$ |
| OPTIMAL_P | $\alpha_{ef} - \dfrac{\alpha_{ep}}{\alpha_{ep}+\alpha_{np}+1}$ |
| Ochiai | $\dfrac{\alpha_{ef}}{\sqrt{\alpha_{ef}+\alpha_{nf}}+\sqrt{\alpha_{ef}+\alpha_{ep}}}$ |
| GP02 | $2(\alpha_{ef} + \sqrt{\alpha_{np}}) + \sqrt{\alpha_{ep}}$ |
| GP03 | $\sqrt{\left|\alpha_{ef}^{2} - \sqrt{\alpha_{ep}}\right|}$ |
| GP19 | $\alpha_{ef}\sqrt{\left|\alpha_{ep} - \alpha_{ef} + \alpha_{nf} - \alpha_{np}\right|}$ |

Table notes Phasellus venenatis, tortor nec vestibulum mattis, massa tortor interdum felis, nec pellentesque metus tortor nec nisl. Ut ornare mauris tellus, vel dapibus arcu suscipit sed.

environment in which our experiments were carried out was on a computer containing an Intel I9-10900K CPU with 128G physical memory. The operating system was Ubuntu 16.04.3. We conducted experiments on Modelsim SE, Visual Studio2019, and MATLAB R2016b.

## Evaluation

To evaluate the effectiveness of Detraque, we adopt two widely used metrics: *EXAM* [27], and relative improvement (referred to as *RImp*) [28]. *EXAM* is defined as the percentage of executable statements to be examined before finding the actual faulty statement; a lower value of *EXAM* indicates better performance. *RImp* compares two fault location approaches to see the improvement of one approach over the other. Given two approaches FL1 and FL2, FL2 is the baseline approach. *RImp* compares the total number of statements that need to be examined to find all faults using FL1 versus the number required for FL2. A lower value of *RImp* indicates that FL1 outperforms FL2. In addition, in order to further investigate the results, it's essential to perform statistical analysis of the data; the statistical method used here is the Wilcoxon-Signed-Rank Test [29].

## Test suite

**Data collection.**   In this paper, we implement the RTL Verilog test suite on the basis of real programs obtained from OpenCores. Different faulty versions of the program are generated by manually injecting faults into the code. At the same time, however, the types of faults in above test suites can hardly reflect the advantages of dynamic execution tracing. As noted above, static analysis techniques cannot adequately detect faults such as arithmetic symbol faults or conditional statement faults; thus we deliberately increased the number of such faults during fault injection.

The programs come from real-world development in OpenCores. As shown in Table 2, the names of the seven groups of programs are AES, Cordic, FPU, SHA256, Pci, Uart and USB. AES, the full name of which is Advanced Encryption Standard, is one of the most popular algorithms in symmetric key encryption. The Cordic algorithm is a digital coordinate rotation calculation method that is mainly used for the calculation of trigonometric functions, hyperbolas, exponents, and logarithms. FPU, or float point unit, is a part of a computer system that is specifically used for floating-point operations. SHA256 comes from the secure hash algorithm, a cryptographic hash function algorithm standard. The common point of these four groups of programs is that they are all computationally intensive. This type of program helps us to insert the types of faults discussed above that may be ignored by static analysis techniques. The other three programs (Pci, Uart, and USB) are all communication interface programs. One feature of this type of program is a larger number of branches and thus more conditional statements to facilitate fault injection. Fault injection in statements related to conditional statements will reduce the difficulty of failure case exposure in test cases. The faulty versions of each program are created by seeding individual faults into the code. Creating *N* faulty versions from the same base program has significant benefits: specifically, the understanding gained from studying the code applies to all *N* versions, and the work involved in generating the test pools is applicable to all versions.

**Fault injection.**   Our fault injection method is inspired by [24]. The faults injected into the RTL source code can be broadly divided into control-dependency faults and data-dependency faults. As mentioned above, it is challenging to locate arithmetic symbol faults or conditional statement faults through static analysis. In order to better reflect Detraque's ability to locate these types of faults, we also screened the injected faults. Table 3 contains several examples of

**Table 2. Test suite information.**

| Program Name | Executable Lines | Faulty Versions | Test Pool | Detection Ratio |
|---|---|---|---|---|
| AES | 528 | 6 | 2663 | 0.0006-0.072 |
| Cordic | 239 | 8 | 2771 | 0.0006-0.095 |
| FPU | 293 | 10 | 1580 | 0.0027-0.131 |
| SHA256 | 553 | 13 | 4119 | 0.0075-0.089 |
| Pci | 957 | 8 | 1552 | 0.0008-0.025 |
| Uart | 318 | 6 | 1061 | 0.0017-0.163 |
| USB | 298 | 10 | 2133 | 0.0027-0.101 |

Table notes Phasellus venenatis, tortor nec vestibulum mattis, massa tortor interdum felis, nec pellentesque metus tortor nec nisl. Ut ornare mauris tellus, vel dapibus arcu suscipit sed.

fault injection; here $C$ denotes control-dependency faults while $D$ represents data-dependency faults.

**Test adequacy criteria.** The existence of the (presumed correct) base version supplies us with an oracle for use in checking the results of test cases executed on the faulty versions. There are two stages in test case generation. The first stage consists of creating a set of test cases in line with good testing practices, based on the developers' understanding of the Verilog programs' functionality, knowledge of special values and boundary points that are easily observable in the code. The second involves manually generating a set of test cases in the test-bench; the latter is a supplement to the former, and the purpose is to meet the test adequacy standards mentioned in [30]. One of the most important indicators is *Detection Ratio* (as shown in Table 2), which refers to the ratio of test cases that can detect fault in a certain faulty version of the program to the total test cases. This indicator can effectively judge the rationality of the injection faults.

## Data analysis

In this subsection, we utilize *EXAM*, *RImp*, and the statistical comparison to evaluate the effectiveness of Detraque.

**EXAM distribution**. Table 4 illustrates the *Exam* score of seven state-of-the-art fault localization techniques (seven suspiciousness evaluation metrics) on seven subject programs. The rightmost column (*Average*) is the average *EXAM* of seven suspiciousness evaluation metrics on subject programs. Taking Russel_Rao on AES as an example, the *EXAM* score is 0.183; this means that before the faulty statements of six faulty versions (see Table 2) are located, 18.3% of executable statements on average have been examined. The rightmost *Average EXAM* score of AES is 0.306; this means that the average *EXAM* score of AES on the seven suspiciousness

**Table 3. Sample of fault injection.**

| Fault Type | Injected Version | Bug Detail |
|---|---|---|
| C | AES.v2 | Change of logical operator |
| D | Cordic.v5 | Changed & to \| causing wrong data assignment |
| D | FPU.v1 | Wrong assignment causing wrong data to |
| C | Uart.v2 | Changed the case condition |

Table notes Phasellus venenatis, tortor nec vestibulum mattis, massa tortor interdum felis, nec pellentesque metus tortor nec nisl. Ut ornare mauris tellus, vel dapibus arcu suscipit sed.

**Table 4. *EXAM* of seven programs on seven evaluation metrics.**

| Program name | *EXAM* of statements' suspiciousness | | | |
|---|---|---|---|---|
|  | **Russel_Rao** | **GP02** | **GP03** | **Dstar** |
| **AES** | 0.183 | 0.466 | 0.267 | 0.183 |
| **Cordic** | 0.332 | 0.284 | 0.429 | 0.332 |
| **FPU** | 0.577 | 0.512 | 0.593 | 0.581 |
| **SHA256** | 0.260 | 0.264 | 0.263 | 0.265 |
| **Pci** | 0.425 | 0.623 | 0.418 | 0.425 |
| **Uart** | 0.655 | 0.557 | 0.635 | 0.655 |
| **USB** | 0.400 | 0.490 | 0.510 | 0.400 |
|  | **OPTIMAL_P** | **GP19** | **Ochiai** | **Average** |
| **AES** | 0.469 | 0.286 | 0.286 | 0.306 |
| **Cordic** | 0.318 | 0.332 | 0.332 | 0.337 |
| **FPU** | 0.609 | 0.591 | 0.591 | 0.579 |
| **SHA256** | 0.270 | 0.262 | 0.262 | 0.264 |
| **Pci** | 0.745 | 0.426 | 0.420 | 0.498 |
| **Uart** | 0.597 | 0.647 | 0.534 | 0.612 |
| **USB** | 0.521 | 0.414 | 0.448 | 0.455 |

Table notes Phasellus venenatis, tortor nec vestibulum mattis, massa tortor interdum felis, nec pellentesque metus tortor nec nisl. Ut ornare mauris tellus, vel dapibus arcu suscipit sed.

evaluation metrics is 0.306. The data in Table 4 shows that the average *EXAM* across all seven programs is 43.6%, indicating that Detraque achieves high Verilog fault localization effectiveness. Although the effect on the programs Uart and FPU is ordinary, its performance on the other groups of programs is good. For developers, especially when facing large-scale Verilog programs, this approach can save a lot of manpower and reduce the material costs of troubleshooting. To the best of our knowledge, this work is among the earliest attempts to apply dynamic execution tracing technology to Verilog fault localization, and is accordingly valuable.

According to subsequent analysis, among the first four computationally intensive programs, *EXAM* averages 37.1%. In the last three control-intensive programs, *EXAM* averages 52.2%. Notably, the difference between these figures is significant. Under the premise that the fault injection types (control-dependency faults and data-dependency faults) of these two types of programs are basically evenly distributed, Detraque is less effective in control-intensive programs. One possible reason is that control-intensive programs contain more conditional statements (such as *if* and *case*). In Verilog, the *if* specifies a priority coding logic, while the logic generated by the *case* is parallel and not prioritized. This lack of priority means that the faults injected in *case*-related conditional statements may not change the statement coverage during program execution, which in turn affects the experimental results of Detraque.

Fig 4 illustrates the *EXAM* distribution of seven evaluation metrics in each group of programs. For each curve, the x-axis represents the percentage of executable statements examined, while the y-axis denotes the percentage of faults already located in all faulty versions. A point in Fig 4 indicates the percentage of faults located after a given percentage of executable statements has been examined in each faulty version. From the seven curves of the figure, there is no obvious difference between the curves corresponding to each evaluation metric, although Dstar (marked with a blue circle) is generally above the other curves. This shows that, under

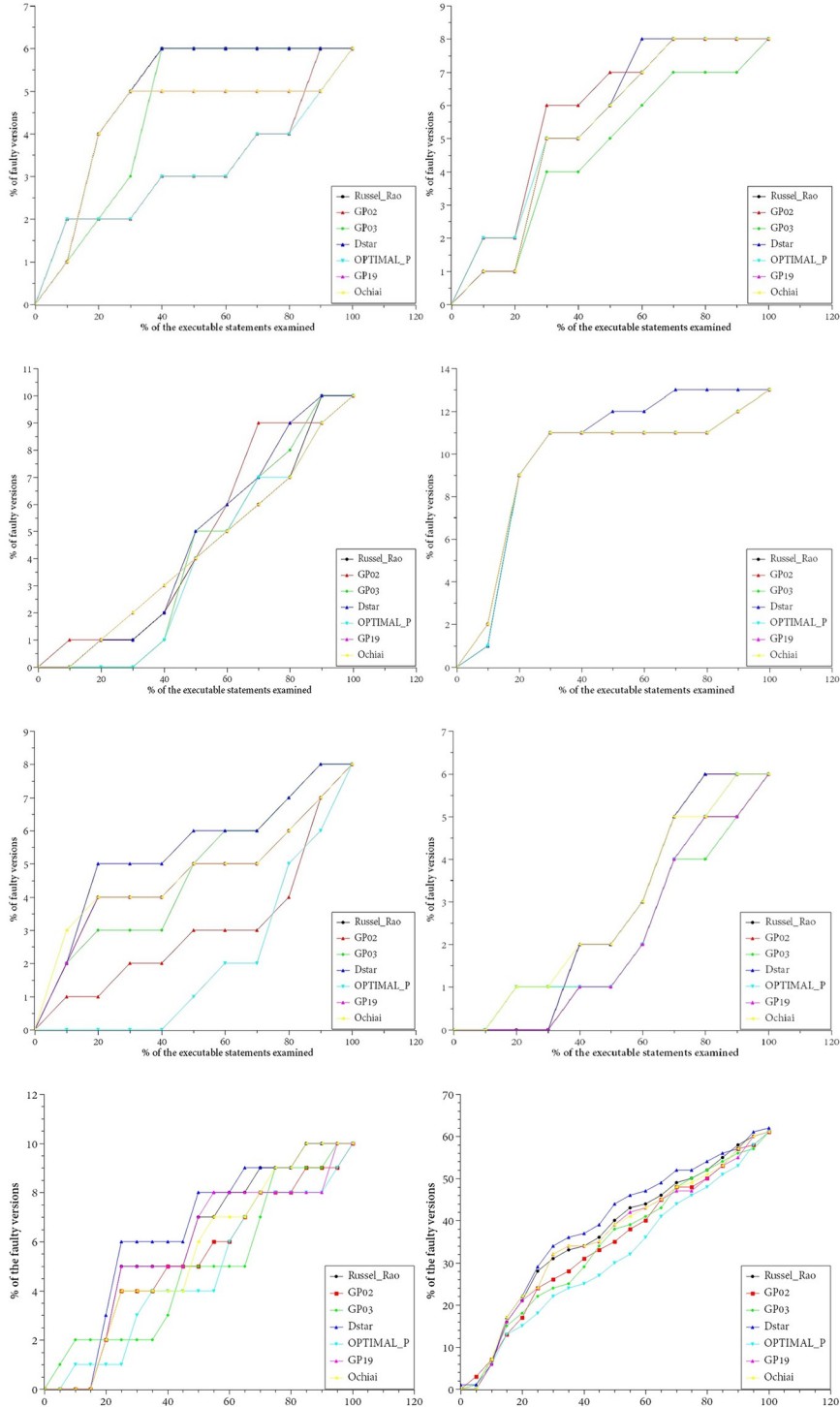

**Fig 4. *EXAM* comparison of seven evaluation metrics.** (a): The *EXAM* Comparison on AES. (b): The *EXAM* Comparison on Cordic. (c): The *EXAM* Comparison on FPU. (d): The *EXAM* Comparison on SHA256. (e): The *EXAM* Comparison on Pci. (f): The *EXAM* Comparison on Uart. (g): The *EXAM* Comparison on USB. (h): The *EXAM* Comparison on all faulty versions.

the *EXAM* evaluation index, there is no significant difference in the fault localization effect of each ranking metric on the test set, and that Dstar performs slightly better.

*RImp* **distribution**. For a more detailed comparison, we use *RImp* to evaluate the seven evaluation metrics. For each group of programs in the test suite, we choose the evaluation metric with the relatively best localization effect as a reference. The test programs in Fig 5(a) to 5 (g) are AES, Cordic, FPU, SHA256, Pci, Uart, and USB, while the evaluation metrics used as reference are Dstar, GP02, GP02, Russel_Rao, GP03, OPTIMAL_P, and GP03. The data in Fig 5(h) is derived from all the faulty versions in the test suite.

In Fig 5(a) to 5(g), compared with the relatively worse evaluation metrics, the statements of the comparatively superior evaluation metric that need to be examined are reduced by an amount ranging from 39% to 98%. The seven relatively superior evaluation metrics are Dstar, GP02, GP02, Russel_Rao, GP03, GP02, and GP03. Taking Fig 5(b) as an example, the best evaluation metric is GP02; compared with GP03, the saving is 33.8% (100%—66.2% = 33.8%). This means that the number of statements checked could be reduced by 33.8%. The maximum saving is 33.8% (100%—66.2% = 33.8%) on GP03, while the minimum saving is 14.5% (100%—85.5% = 14.5%) on Russel_Rao, Dstar, GP19, and Ochiai. Fig 5(h) shows that for all the faulty versions of the seven subject programs, Dstar performs best.

**Statistical comparison**. From Fig 5(a) to 5(h), it is shown that significant differences exist between the different evaluation metrics. In order to further qualitatively analyze the experimental results, it's necessary to use a statistical analysis method, as shown in Table 5. We select the Wilcoxon-Signed-Rank Test [29] to achieve this goal. The Wilcoxon-Signed-Rank Test is a non-parametric statistical hypothesis test for testing the differences between pairs of measurements $F(x)$ and $G(y)$. The experiments involved performing one paired Wilcoxon-Signed-Rank test between each two localization models by using *EXAM* for the pairs of measurements $F(x)$ and $G(y)$. Specifically, each test uses both the 2-tailed and 1-tailed testing at an $\sigma$ level of 0.05. It's obvious that Dstar obtains the best result on AES and Uart, GP02 obtains the best result on Cordic and FPU, Russel_Rao obtains the best result on SHA256, GP03 obtains the best result on Pci and Ochiai obtains the best result on Uart. Furthermore, referring to the rightmost column of Table 5, the Dstar obtains "BETTER" results relative to the others for all faulty versions. Accordingly, the Dstar performs significantly better.

## Discussion

In section Experimental Study, we present an empirical evaluation of Detraque on our benchmark suite of hardware defect scenarios. In this section, We address the following research questions to discuss:

**RQ1**. How effective is Detraque at guiding the process for fault localization to a circuit description?

**RQ2**. Does Detraque perform better at arithmetic symbol fault or conditional statement fault localization compared to model-based method?

**RQ3**. Is the different performance of different suspicious value evaluation metrics related to the language characteristics of HDL?

### RQ1. Effectiveness analysis

Experimental results show that Detraque achieve the average *EXAM* of 43.6% on all faulty versions in the test suite proposed in this paper. Any degree of scope reduction is helpful for fault localization of large-scale programs. In order to further verify the experimental effect, we also

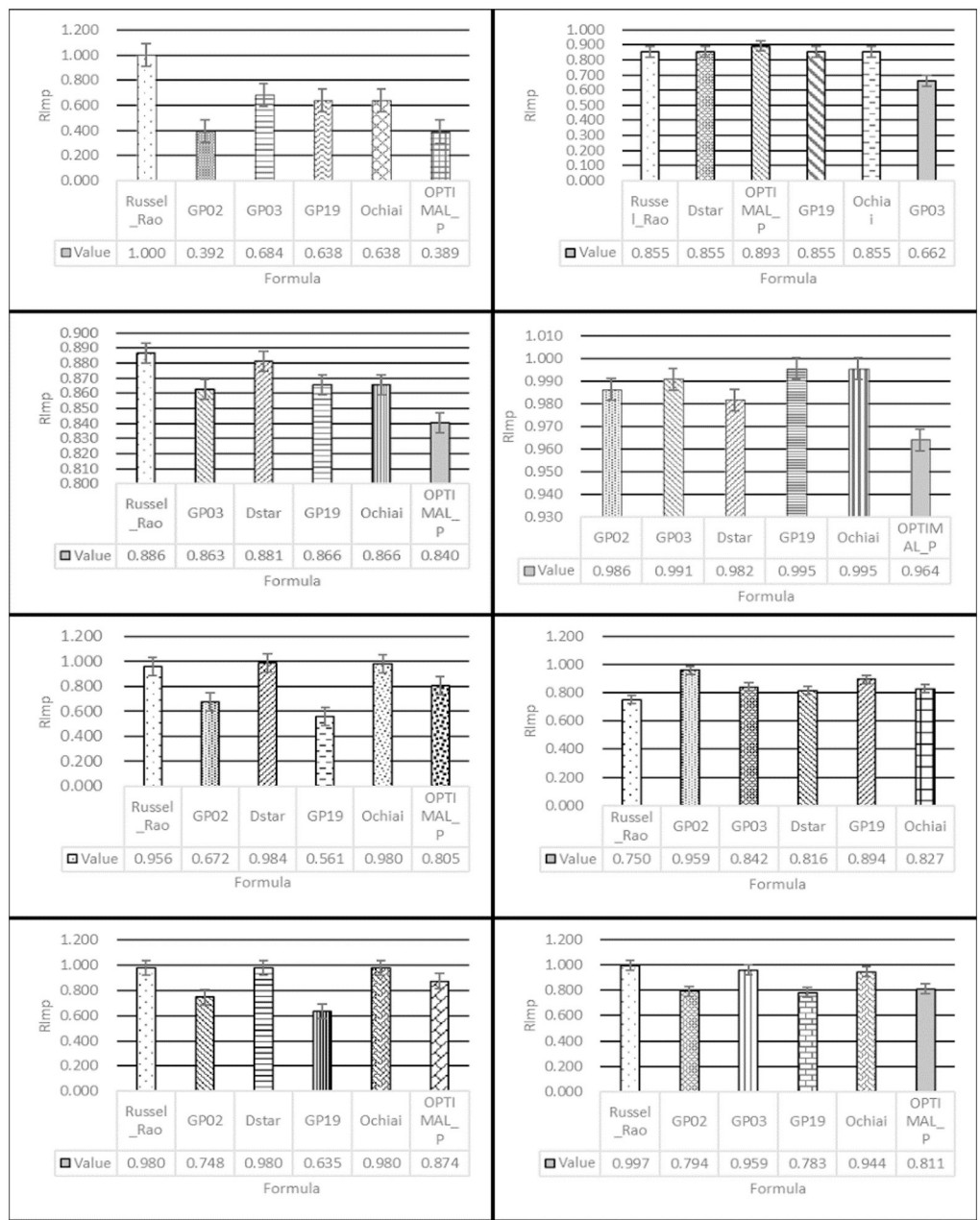

**Fig 5. *RImp* comparison of remaining evaluation metrics.** (a): The *RImp* of Dstar over others on AES. (b): The *RImp* of Dstar over others on Cordic. (c): The *RImp* of Dstar over others on FPU. (d): The *RImp* of Dstar over others on SHA256. (e): The *RImp* of Dstar over others on Pci. (f): The *RImp* of Dstar over others on Uart. (g): The *RImp* of Dstar over others on USB. (h): The *RImp* of Dstar over others on all faulty versions.

conducted experiments on the traditional 74XXX/ISCAS85 benchmark suite of logic circuits. The same seven metrics are selected to evaluate suspicion value. The average on *EXAM* is 40.2%. It shows that, as a supplement to traditional static analysis methods, Detraque is effective on both test suites.

**Table 5. Wilcoxon-Signed-Rank test of Detraque on seven programs.**

| Program | Evaluation Metrics | 2-tailed | 1-tailed(right) | 1-tailed(left) | Conclusion |
|---|---|---|---|---|---|
| AES | Dstar vs Russel_Rao | 1.00e+00 | 1.00e+00 | 1.00e+00 | SIMILAR |
| | Dstar vs GP02 | 2.21e-01 | 9.13e-01 | 1.38e-01 | BETTER |
| | Dstar vs GP03 | 6.84e-01 | 7.06e-01 | 3.93e-01 | BETTER |
| | Dstar vs GP19 | 2.21e-01 | 9.13e-01 | 1.38e-01 | BETTER |
| | Dstar vs Ochiai | 3.17e-01 | 9.77e-01 | 5.00e-01 | BETTER |
| | Dstar vs OPTIMAL_P | 3.17e-01 | 9.77e-01 | 5.00e-01 | BETTER |
| | Total | 3.12e-05 | 9.85e-01 | 1.66e-5 | BETTER |
| Cordic | GP02 vs Russel_Rao | 1.79e-01 | 9.63e-01 | 1.86e-01 | BETTER |
| | GP02 vs GP03 | 1.79e-01 | 9.63e-01 | 1.86e-01 | BETTER |
| | GP02 vs Dstar | 1.79e-01 | 9.63e-01 | 1.86e-01 | BETTER |
| | GP02 vs OPTIMAL_P | 3.17e-01 | 9.77e-01 | 5.00e-01 | BETTER |
| | GP02 vs GP19 | 1.79e-01 | 9.63e-01 | 1.86e-01 | BETTER |
| | GP02 vs Ochiai | 1.79e-01 | 9.63e-01 | 1.86e-01 | BETTER |
| | Total | 1.83e-05 | 9.91e-01 | 9.39e-6 | BETTER |
| FPU | GP02 vs Russel_Rao | 7.96e-02 | 9.70e-01 | 5.28e-02 | BETTER |
| | GP02 vs GP03 | 1.15e-01 | 9.53e-01 | 7.11e-02 | BETTER |
| | GP02 vs Dstar | 1.15e-01 | 9.53e-01 | 7.11e-02 | BETTER |
| | GP02 vs OPTIMAL_P | 2.48e-01 | 8.95e-01 | 1.47e-01 | BETTER |
| | GP02 vs GP19 | 2.02e-01 | 9.07e-01 | 1.10e-01 | BETTER |
| | GP02 vs Ochiai | 2.02e-01 | 9.07e-01 | 1.10e-01 | BETTER |
| | Total | 3.72e-05 | 9.81e-01 | 1.92e-05 | BETTER |
| SHA256 | Russel_Rao vs GP02 | 5.92e-01 | 7.88e-01 | 3.94e-01 | BETTER |
| | Russel_Rao vs GP03 | 6.54e-01 | 8.14e-01 | 5.00e-01 | BETTER |
| | Russel_Rao vs Dstar | 2.85e-01 | 9.09e-01 | 2.11e-01 | BETTER |
| | Russel_Rao vs OPTIMAL_P | 1.08e-01 | 9.69e-01 | 9.07e-02 | BETTER |
| | Russel_Rao vs GP19 | 5.92e-01 | 7.88e-01 | 3.94e-01 | BETTER |
| | Russel_Rao vs Ochiai | 5.92e-01 | 7.88e-01 | 3.94e-01 | BETTER |
| | Total | 2.03e-04 | 9.99e-01 | 1.04e-04 | BETTER |
| Pci | GP03 vs Russel_Rao | 8.92e-01 | 6.06e-01 | 5.00e-01 | BETTER |
| | GP03 vs GP02 | 2.48e-01 | 8.95e-01 | 1.47e-01 | SIMILAR |
| | GP03 vs Dstar | 8.92e-01 | 6.06e-01 | 5.00e-01 | BETTER |
| | GP03 vs OPTIMAL_P | 4.63e-02 | 9.81e-01 | 2.95e-02 | BETTER |
| | GP03 vs GP19 | 6.12e-01 | 7.22e-01 | 3.36e-01 | BETTER |
| | GP03 vs Ochiai | 1.00e+00 | 5.72e-01 | 5.72e-01 | SIMILAR |
| | Total | 2.83e-03 | 9.86e-01 | 1.46e-04 | BETTER |
| Uart | Ochiai vs Russel_Rao | 1.79e-01 | 9.63e-01 | 1.85e-01 | BETTER |
| | Ochiai vs GP02 | 1.00e+00 | 6.05e-01 | 6.05e-01 | SIMILAR |
| | Ochiai vs GP03 | 5.92e-01 | 7.88e-01 | 3.94e-01 | BETTER |
| | Ochiai vs Dstar | 1.79e-01 | 9.63e-01 | 1.85e-01 | BETTER |
| | Ochiai vs OPTIMAL_P | 3.17e-01 | 9.77e-01 | 5.00e-01 | BETTER |
| | Ochiai vs GP19 | 2.85e-01 | 9.09e-01 | 2.11e-01 | SIMILAR |
| | Total | 3.72e-04 | 9.81e-01 | 1.92e-04 | BETTER |

(*Continued*)

**Table 5.** (Continued)

| Program | Evaluation Metrics | 2-tailed | 1-tailed(right) | 1-tailed(left) | Conclusion |
|---|---|---|---|---|---|
| USB | Dstar vs Russel_Rao | 1.00e+00 | 1.00e+00 | 1.00e+00 | SIMILAR |
| | Dstar vs GP02 | 4.76e-01 | 7.79e-01 | 2.57e-01 | BETTER |
| | Dstar vs GP03 | 3.73e-01 | 8.28e-01 | 2.03e-01 | BETTER |
| | Dstar vs OPTIMAL_P | 3.74e-01 | 8.28e-01 | 2.03e-01 | BETTER |
| | Dstar vs GP19 | 3.17e-01 | 9.77e-01 | 5.00e-01 | BETTER |
| | Dstar vs Ochiai | 4.14e-01 | 8.61e-01 | 2.93e-01 | SIMILAR |
| | Total | 7.64e-04 | 9.93e-01 | 1.78e-04 | BETTER |
| All versions | Dstar vs Russel_Rao | 3.73e-01 | 8.28e-01 | 2.03e-01 | BETTER |
| | Dstar vs GP02 | 2.45e-01 | 9.19e-01 | 2.17e-01 | BETTER |
| | Dstar vs GP03 | 6.12e-01 | 7.88e-01 | 3.75e-01 | BETTER |
| | Dstar vs GP19 | 1.79e-01 | 9.63e-01 | 1.85e-01 | BETTER |
| | Dstar vs Ochiai | 4.76e-01 | 7.79e-01 | 2.57e-01 | BETTER |
| | Dstar vs OPTIMAL_P | 2.85e-01 | 9.09e-01 | 2.11e-01 | BETTER |
| | Total | 1.64e-06 | 9.96e-01 | 9.67e-06 | BETTER |

Table notes Phasellus venenatis, tortor nec vestibulum mattis, massa tortor interdum felis, nec pellentesque metus tortor nec nisl. Ut ornare mauris tellus, vel dapibus arcu suscipit sed.

## RQ2. Performance for specific fault

In the section Introduction, it is mentioned that traditional static analysis methods are difficult to detect for the two types of fault, arithmetic symbol fault and conditional statement fault. The fault injection of the test suite in this paper is mainly based on these two types of faults. Experimental results show that Detraque is effective on our test suite. To make the effectiveness of Detraque on specific fault more convincing, we add an experiment. We include results for GDE, a state-of-the-art model-based engine [31]. We use *EXAM* as the measurement, and the statistical results based on Wilcoxon-Signed-Rank Test are shown in Table 6. The statistical results show that Detraque performs better on the seven programs. In the test suite dominated by the two types of faults (arithmetic symbol or conditional statement faults) proposed in this paper, our method is better than GDE.

**Table 6. Wilcoxon-Signed-Rank test of Detraque vs GDE.**

| Program | 2-tailed | 1-tailed (right) | 1-tailed (left) | Conclusion |
|---|---|---|---|---|
| AES | 2.57e-01 | 7.87e-01 | 1.78e-01 | BETTER |
| Cordic | 3.17e-01 | 2.57e-01 | 1.78e-01 | BETTER |
| FPU | 1.15e-01 | 1.78e-01 | 1.78e-01 | BETTER |
| SHA256 | 5.92e-01 | 4.32e-01 | 2.03e-01 | BETTER |
| Pci | 1.79e-01 | 1.83e-01 | 2.51e-01 | BETTER |
| Uart | 1.82e-01 | 6.12e-01 | 1.79e-01 | BETTER |
| USB | 4.77e-01 | 3.17e-01 | 3.94e-01 | BETTER |
| All versions | 9.09e-01 | 2.01e-01 | 2.17e-01 | BETTER |

Table notes Phasellus venenatis, tortor nec vestibulum mattis, massa tortor interdum felis, nec pellentesque metus tortor nec nisl. Ut ornare mauris tellus, vel dapibus arcu suscipit sed.

### RQ3. Why does Dstar perform better?

The principle of Dstar is the contribution of the first failed test case that executes it in computing its likelihood of containing a bug is larger than or equal to that of the second failed test case that executes it, which in turn is larger than or equal to that of the third failed test case that executes it, and so on. This principle has been verified to be correct on *Siemens suite* [32] and *Unix suite* [33]. The performance of Dstar in the test suite of this paper shows to some extent the effectiveness of this principle on Verilog. Whether or not it is a determinant of better Dstar performance requires further research.

## Limitations and threats to validate

Our experimental results suggest that Detraque is effective at automatically locating faults in Verilog. However, there are several limitations to our approach and threats to the validity.

### Timing bugs

Faults in HDL descriptions stemming from timing flow issues and incorrect circuit behavior with respect to the clock signal often go undetected by a traditional test-bench, requiring complicated analyses of waveforms from the simulation to be identified. Such timing bugs are therefore not within the scope of our approach, which relies heavily on testbenches to assess the functional correctness of designs. We note that while such bugs are complex to debug, they represent only a subset of hardware defects in industry, while a non-trivial amount of defects in hardware correspond to functional correctness [34].

### Threats to validity

The main purpose of this work is to verify the effectiveness of Detraque in Verilog fault localization and to compare different evaluation metrics. No comparison is conducted with the current mainstream model-based HDLs fault localization methods. This will be the focus of our next phase of work.

This paper also assumes perfect bug detection which may not always hold in practice. Our fault localization methods provide a programmer with a ranked list of statements sorted in order of their suspiciousness which may then be examined in order to identify the faults in the statements. We assume that if a programmer examines a faulty statement, the programmer shall consequently also identify the corresponding fault. Additionally, the programmer shall not identify a non-faulty statement incorrectly as faulty. Should such perfect bug detection not hold, then the amount of code that needs to be examined to detect the bug may increase. However, such a concern applies equally to other fault localization methods.

## Conclusion and future work

This paper aims primarily to explore Verilog fault localization technology. First, we built a relatively complete RTL Verilog test suite based on Verilog programs in actual applications. Then Detraque is proposed, which is the first work to apply dynamic execution tracking technology to Verilog fault localization. After collecting the dynamic execution information of the program, Detraque calculates the suspicion value of the statements and outputs a ranking list in descending order of suspiciousness. Furthermore, we compared different evaluation metrics in terms of *EXAM* and *RImp*. The experimental results show that the proposed Detraque is effective in locating arithmetic symbol or conditional statement faults in Verilog, and could accordingly be considered as a supplement to the static analysis method. Moreover, it is

experimentally verified that when Detraque uses the Dstar suspicion value evaluation metric, it achieves the best performance on the test suite.

In the future, more fault localization techniques will be tested on Verilog, such as slice-based techniques and machine learning-based techniques. Our research team will further explore suitable suspicion value metrics for Verilog in combination with the language characteristic of Verilog. And we will further explore the construction of the RTL Verilog test suite to ensure that it meets the needs of fault localization research and lay the foundation for long-term research in the future.

## Author Contributions

**Conceptualization:** Jiang Wu, Jianjun Xu, Xiaoguang Mao.

**Data curation:** Jiang Wu, Jianjun Xu, Jiayu He, Xiankai Meng.

**Formal analysis:** Jiang Wu, Xiankai Meng.

**Funding acquisition:** Jiang Wu, Jianjun Xu, Xiaoguang Mao.

**Investigation:** Jiang Wu, Zhuo Zhang, Jiayu He.

**Methodology:** Jiang Wu, Zhuo Zhang, Jiayu He.

**Project administration:** Jiang Wu, Zhuo Zhang.

**Resources:** Jiang Wu.

**Software:** Jiang Wu, Jiayu He, Panpan Li.

**Supervision:** Jiang Wu.

**Validation:** Jiang Wu.

**Visualization:** Jiang Wu.

**Writing – original draft:** Jiang Wu.

**Writing – review & editing:** Jiang Wu, Panpan Li.

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
