## [Decision Letter · Decision Letter 0]

15 Jun 2022

PONE-D-22-09808Detraque: Dynamic Execution Tracing Techniques for Automatic Fault Localization of Hardware Design CodePLOS ONE

Dear Dr. Wu,

Thank you for submitting your manuscript to PLOS ONE. After careful consideration, we feel that it has merit but does not fully meet PLOS ONE’s publication criteria as it currently stands. Therefore, we invite you to submit a revised version of the manuscript that addresses the points raised during the review process.All the comments of the reviewers must be addressed and necessary modifications must be done on the revise manuscript.Please submit your revised manuscript by Jul 30 2022 11:59PM. If you will need more time than this to complete your revisions, please reply to this message or contact the journal office at plosone@plos.org. Please include the following items when submitting your revised manuscript:A rebuttal letter that responds to each point raised by the academic editor and reviewer(s). You should upload this letter as a separate file labeled 'Response to Reviewers'.A marked-up copy of your manuscript that highlights changes made to the original version. You should upload this as a separate file labeled 'Revised Manuscript with Track Changes'.An unmarked version of your revised paper without tracked changes. You should upload this as a separate file labeled 'Manuscript'.If applicable, we recommend that you deposit your laboratory protocols in protocols.io to enhance the reproducibility of your results. Protocols.io assigns your protocol its own identifier (DOI) so that it can be cited independently in the future. For instructions see: https://journals.plos.org/plosone/s/submission-guidelines#loc-laboratory-protocols. Additionally, PLOS ONE offers an option for publishing peer-reviewed Lab Protocol articles, which describe protocols hosted on protocols.io. Read more information on sharing protocols at https://plos.org/protocols?utm_medium=editorial-email&utm_source=authorletters&utm_campaign=protocols.

We look forward to receiving your revised manuscript.

Kind regards,

Lalit Chandra Saikia, PhD

Academic Editor

PLOS ONE

Journal Requirements:

Additional Editor Comments:

All the comments of the reviewers must be addressed and accordingly necessary modification must be done in the revised manuscript.

Reviewers' comments:

Reviewer's Responses to Questions

**Comments to the Author**

1. Is the manuscript technically sound, and do the data support the conclusions?

Reviewer #1: Yes

Reviewer #2: Yes

2. Has the statistical analysis been performed appropriately and rigorously? 

Reviewer #1: Yes

Reviewer #2: Yes

3. Have the authors made all data underlying the findings in their manuscript fully available?

Reviewer #1: Yes

Reviewer #2: Yes

4. Is the manuscript presented in an intelligible fashion and written in standard English?

Reviewer #1: Yes

Reviewer #2: Yes

5. Review Comments to the Author

Reviewer #1: 1) Please review your English grammar. For instance, "we" have been used many times.

2) All the figures are not available in the text, and they all are available in the end which is difficult to follow.

3) Please check the format of your references. For instance, Reference 32 has capital letters which is different than others. You have used "//". Please check again.

4) Please check your table format. For example, Table 3 in page number 10.

Reviewer #2: It is a very valuable contribution. This will be more beneficial if some related material from previous issues of the journal in the literature review will be added, however,the literature review has done rigorously.

6. PLOS authors have the option to publish the peer review history of their article (what does this mean?). If published, this will include your full peer review and any attached files.

Reviewer #1: **Yes: **Krishna Mohan Kudiri

Reviewer #2: No

---

## [Author Response · Author response to Decision Letter 0]

21 Jun 2022

Dear Editors and Reviewers,

We would like to thank you and the anonymous reviewers for their constructive comments and recommendations to improve the manuscript. We have addressed all the comments. The details of our response are below. We are happy to answer any further questions that might arise.

Yours Sincerely,

Jiang Wu, Zhuo Zhang, Jianjun Xu, Jiayu He, Xiaoguang Mao, Xiankai Meng, and Panpan Li

Response to Journal Requirements:

Comment: Upon re-submitting your revised manuscript, please upload your study’s minimal underlying data set as either Supporting Information files or to a stable, public repository and include the relevant URLs, DOIs, or accession numbers within your revised cover letter.

Response: We have upload our study’s minimal underlying data set to https://github.com/wndif/Benchmark_Detraque.git. We also included the Github link on Page 3 of the article.

1. Response to Reviewer #1’s comments:

1.1 Comment: 

Please review your English grammar. For instance, "we" have been used many times.

Response: We very much appreciate your constructive recommendations. In summary, we have made the following major changes according to your advice in this revision: we first passed a manual grammar check and then used the tool “Grammarly” to further check the grammar. At the same time, we also replaced a large number of "we" with more appropriate pronouns.

1.2 Comment: 

All the figures are not available in the text, and they all are available in the end which is difficult to follow.

Response: Sorry for making it difficult for you to read. Separation of paper text and figures is a submission requirement of PLOS ONE, so figures are attached at the end of the text when submitting. It was merged together in the final published version. Meanwhile, we have also adjusted the position of the figures in the text and further marked the reference to the figures in the text, so as to help readers better grasp the content of the figures.

1.3 Comment: 

Please check the format of your references. For instance, Reference 32 has capital letters which is different than others. You have used "//". Please check again?

Response: We very much appreciate your constructive comments and sorry for our inappropriate illustration. We have revised your problem with “Reference 32” and "//" on Section Reference. We then carefully checked the format of all references and refined them.

1.4 Comment: 

Please check your table format. For example, Table 3 in page number 10.?

Response: Thank you for the advice. We have revised the “Table 3 in page number 10”. We then carefully checked the format of all tables and refined them.

2. Response to Reviewer #2’s comments:

2.1 Comment: 

It is a very valuable contribution. This will be more beneficial if some related material from previous issues of the journal in the literature review will be added, however, the literature review has done rigorously.

Response: We very much appreciate your constructive comments and sorry for our inappropriate illustration. According to your advise, we have added two articles from previous issues of the journal and noted them in the revised version, they are:

1.Qu N, You W. Design and fault diagnosis of DCS sintering furnace's temperature control system for edge computing[J]. PloS one, 2021, 16(7): e0253246.

2. Xiao Y, Wang K, Liu W, et al. Research on rapier loom fault system based on cloud-side collaboration[J]. PloS one, 2021, 16(12): e0260888.

---

## [Decision Letter · Decision Letter 1]

30 Aug 2022

Detraque: Dynamic Execution Tracing Techniques for Automatic Fault Localization of Hardware Design Code

PONE-D-22-09808R1

Dear Dr. Wu,

We’re pleased to inform you that your manuscript has been judged scientifically suitable for publication and will be formally accepted for publication once it meets all outstanding technical requirements.

Kind regards,

Lalit Chandra Saikia, PhD

Academic Editor

PLOS ONE

Additional Editor Comments (optional):

Reviewers' comments:

Reviewer's Responses to Questions

**Comments to the Author**

1. If the authors have adequately addressed your comments raised in a previous round of review and you feel that this manuscript is now acceptable for publication, you may indicate that here to bypass the “Comments to the Author” section, enter your conflict of interest statement in the “Confidential to Editor” section, and submit your "Accept" recommendation.

Reviewer #1: All comments have been addressed

Reviewer #2: All comments have been addressed

2. Is the manuscript technically sound, and do the data support the conclusions?

Reviewer #1: Yes

Reviewer #2: Yes

3. Has the statistical analysis been performed appropriately and rigorously? 

Reviewer #1: Yes

Reviewer #2: (No Response)

4. Have the authors made all data underlying the findings in their manuscript fully available?

Reviewer #1: Yes

Reviewer #2: Yes

5. Is the manuscript presented in an intelligible fashion and written in standard English?

Reviewer #1: Yes

Reviewer #2: Yes

6. Review Comments to the Author

Reviewer #1: 1) Please check the grammar again.

Other than that, this paper is well written. The author addressed all the comments.

Reviewer #2: Authors have addressed all the comments and now paper is according to the reviewer's comments/reviews.

7. PLOS authors have the option to publish the peer review history of their article (what does this mean?). If published, this will include your full peer review and any attached files.

Reviewer #1: **Yes: **Krishna Mohan Kudiri

Reviewer #2: No

---

## [Editor Report · Acceptance letter]

8 Sep 2022

PONE-D-22-09808R1 

Detraque: Dynamic Execution Tracing Techniques for Automatic Fault Localization of Hardware Design Code 

Dear Dr. Wu:

I'm pleased to inform you that your manuscript has been deemed suitable for publication in PLOS ONE. Congratulations! Your manuscript is now with our production department. 

Kind regards, 

on behalf of

Dr. Lalit Chandra Saikia 

Academic Editor

PLOS ONE